# A Neural Network MCMC Sampler that Maximizes Proposal Entropy

## Abstract

Markov Chain Monte Carlo (MCMC) methods sample from unnormalized probability distributions and offer guarantees of exact sampling. However, in the continuous case, unfavorable geometry of the target distribution can greatly limit the efficiency of MCMC methods. Augmenting samplers with neural networks can potentially improve their efficiency. Previous neural network based samplers were trained with objectives that either did not explicitly encourage exploration, or used a L2 jump objective which could only be applied to well structured distributions. Thus it seems promising to instead maximize the proposal entropy for adapting the proposal to distributions of any shape. To allow direct optimization of the proposal entropy, we propose a neural network MCMC sampler that has a flexible and tractable proposal distribution. Specifically, our network architecture utilizes the gradient of the target distribution for generating proposals. Our model achieves significantly higher efficiency than previous neural network MCMC techniques in a variety of sampling tasks. Further, the sampler is applied on training of a convergent energy-based model of natural images. The learned sampler achieves significantly higher proposal entropy and sample quality compared to Langevin dynamics sampler.

## 1 Introduction

Sampling from unnormalized distributions is important for many applications, including statistics, simulations of physical systems, and machine learning. However, the inefficiency of state-of-the-art sampling methods remains a main bottleneck for many challenging applications, such as protein folding (Noé et al., 2019), energy-based model training (Nijkamp et al., 2019), etc.

A prominent strategy for sampling is the Markov Chain Monte Carlo (MCMC) method (Neal, 1993). In MCMC, one chooses a transition kernel that leaves the target distribution invariant and constructs a Markov Chain by applying the kernel repeatedly. The MCMC method relies only on the ergodicity assumption, other than that it is general. If enough computation is performed, the Markov chain generates correct samples from any target distribution, no matter how complex the distribution is. However, the performance of MCMC depends critically on how well the chosen transition kernel explores the state space of the problem. If exploration is ineffective, samples will be highly correlated and of very limited use for downstream applications. Despite some favorable theoretical argument on the effectiveness of some MCMC algorithms, practical implementation of them may still suffer from inefficiencies.

Take, for example, the Hamiltonian Monte Carlo (HMC)(Neal et al., 2011) algorithm, a type of MCMC technique. HMC is regarded state-of-the-art for sampling in continuous spaces Radivojević & Akhmatskaya (2020). It uses a set of auxiliary momentum variables and generates new samples by simulating a Hamiltonian dynamics starting from the previous sample. This allows the sample to travel in state space much further than possible with other techniques, most of whom have more pronounced random walk behavior. Theoretical analysis shows that the cost of traversing a $d$-dimensional state space and generating an uncorrelated proposal is $O(d^{\frac{1}{4}})$ for HMC, which is lower than $O(d^{\frac{1}{3}})$ for Langevine Monte Carlo, and $O(d)$ for random walk. However, unfavorable geometry of a target distribution may still cause HMC to be ineffective because the Hamiltonian dynamics has to be simulated numerically. Numerical errors in the simulation are commonly corrected by a Metropolis-Hastings (MH) accept-reject step for a proposed sample. If the the target

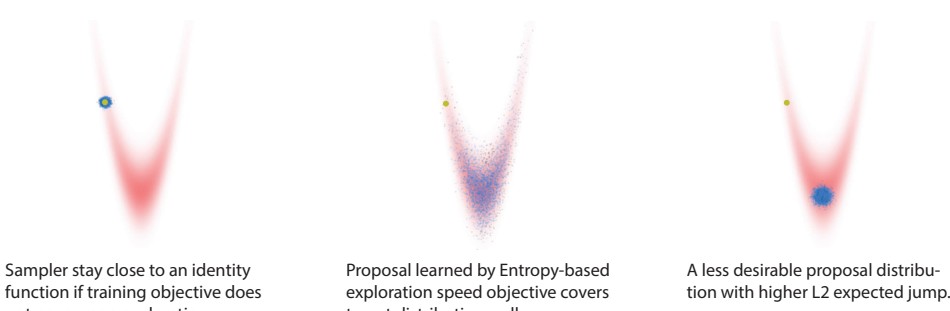

Sampler stay close to an identity function if training objective does not encourage exploration

Proposal learned by Entropy-based exploration speed objective covers target distribution well.

A less desirable proposal distribution with higher L2 expected jump.

Figure 1: Illustration of learning to explore a state space. Larger yellow dot in top left is the initial point $x$, blue and black dots are accepted and rejected samples from the proposal distribution $q(x'|x)$. Solution obtained from optimizing entropy objective is close to the target distribution $p(x)$. However, we can easily construct a less desirable solution with higher L2 jump.

distribution has unfavorable geometric properties, for example, very different variances along different directions, the numerical integrator in HMC will have high error, leading to a very low accept probability (Betancourt et al., 2017). For simple distributions this inefficiency can be mitigated by an adaptive re-scaling matrix (Neal et al., 2011). For analytically tractable distributions, one can also use the Riemann manifold HMC method (Girolami & Calderhead, 2011). But in most other cases, the Hessian required in Riemann manifold HMC algorithm is often intractable or expensive to compute, preventing its application.

Recently, approaches have been proposed that possess the exact sampling property of the MCMC method, while potentially mitigating the described issues with unfavorable geometry. Such approaches include MCMC samplers augmented with neural networks (Song et al., 2017; Levy et al., 2018; Gu et al., 2019), and neural transport MCMC techniques (Hoffman et al., 2019; Nijkamp et al., 2020). A disadvantage of these recent techniques is that their objectives optimize the quality of proposed samples, but do not explicitly encourage exploration speed of the sampler. One notable exception is L2HMC (Levy et al., 2018), a method whose objective includes the size of the expected L2 jump, thereby encouraging exploration. But the L2 expected jump objective is not very general, it only works for simple distributions (see Figure 1, and below).

Another recent work (Titsias & Dellaportas, 2019) proposed a quite general objective to encourage exploration speed by maximizing the entropy of the proposal distribution. In continuous space, the entropy of a distribution is essentially the logarithm of its volume in state space. Thus, the entropy objective naturally encourages the proposal distribution to "fill up" the target state space as well as possible, independent of the geometry of the target distribution. The authors demonstrated the effectiveness of this objective on samplers with simple linear adaptive parameters.

Here we employ the entropy-based objective in a neural network MCMC sampler for optimizing exploration speed. To build the model, we design a flexible proposal distribution for which the optimization of the entropy objective is tractable. Inspired by the HMC algorithm, the proposed sampler uses special architecture that utilizes the gradient of the target distribution to aid sampling. For a 2-D distribution the behavior of the proposed model is illustrated in Figure 1. The sampler, trained with the entropy-based objective, generates samples that explore the target distribution quite well, while it is simple to construct a proposal with higher L2 expected jump (right panel). Later we show the newly proposed method achieves significant improvement in sampling efficiency compared to previous techniques, we then apply the method to the training of an energy-based image model.

## 2 PRELIMINARY: MCMC METHODS, FROM VANILLA TO LEARNED

Consider the problem of sampling from a target distribution $p(x) = e^{-U(x)}/Z$ defined by the energy function $U(x)$ in a continuous state space. MCMC methods solve the problem by constructing and running a Markov Chain, with transition probability $p(x'|x)$, that leaves $p(x)$ invariant. The most general invariance condition is: $p(x') = \int p(x'|x)p(x)dx$ for all $x'$, which is typically enforced by the simpler but more stringent condition of *detailed balance*: $p(x)p(x'|x) = p(x')p(x|x')$.

For a general distribution $p(x)$ it is difficult to directly construct $p(x'|x)$ that satisfies detailed balance, but one can easily[1] make any transition probability satisfy it by including an additional Metropolis-Hastings accept-reject step (Hastings, 1970). When we sample $x'$ at step $t$ from an arbitrary proposal distribution $q(x'|x^t)$, the M-H accept-reject process accepts the new sample $x^{t+1} = x'$ with probability $A(x', x) = \min\left(1, \frac{p(x')q(x^t|x')}{p(x^t)q(x'|x^t)}\right)$. If $x'$ is rejected, the new sample is set to the previous state $x^{t+1} = x^t$. This transition kernel $p(x'|x)$ constructed from $q(x'|x)$ and $A(x', x)$ leaves any target distribution $p(x)$ invariant.

The most popular MCMC techniques use the described M-H accept-reject step to enforce detailed balance, for example, Random Walk Metropolis (RWM), Metropolis-Adjusted Langevin Algorithm (MALA) and Hamiltonian Monte Carlo (HMC). For brevity, we will focus on MCMC methods that use the M-H step, although some alternatives do exist (Sohl-Dickstein et al., 2014). All these methods share the requirement that the accept probability in the M-H step must be tractable to compute. For two of the mentioned MCMC methods this is indeed the case. In the Gaussian random-walk sampler, the proposal distribution is a Gaussian around the current position: $x' = x + \epsilon * \mathcal{N}(0, \mathbf{I})$, which has the form $x' = x + z$. Thus, forward and reverse proposal probabilities are given by $q(x'|x) = p_\mathcal{N}[(x' - x)/\epsilon]$ and $q(x|x') = p_\mathcal{N}[-(x' - x)/\epsilon]$, where $p_\mathcal{N}$ denote the density function of Gaussian. The probability ratio $\frac{q(x^t|x')}{q(x'|x^t)}$ used in the M-H step is therefore equal to 1. In MALA the proposal distribution is a single step of Langevin dynamics with step size $\epsilon$: $x' = x + z$ with $z = -\frac{\epsilon^2}{2}\partial_x U(x) + \epsilon N(0, \mathbf{I})$. We then have $q(x'|x) = p_\mathcal{N}\left[(x' - x)/\epsilon + \frac{\epsilon}{2}\partial_x U(x)\right]$ and $q(x|x') = p_\mathcal{N}\left[-(x' - x)/\epsilon + \frac{\epsilon}{2}\partial_{x'} U(x')\right]$. Both, the forward and reverse proposal probability are tractable since they are the density of Gaussians evaluated at a known location.

Next we show how the HMC sampler can also be formulated as a M-H sampler. Basic HMC involves a Gaussian auxiliary variable $v$ of the same dimension as $x$, which plays the role of the momentum in Physics. HMC sampling consists of two steps: 1. The momentum is sampled from a normal distribution $\mathcal{N}(v; 0, \mathbf{I})$. 2. The Hamiltonian dynamics is simulated for a certain duration with initial condition $x$ and $v$, typically by running a few steps of the leapfrog integrator. Then, a M-H accept-reject process with accept probability $A(x', v', x, v) = \min\left(1, \frac{p(x', v')q(x, v|x', v')}{p(x, v)q(x', v'|x, v)}\right) = \min\left(1, \frac{p(x')p_\mathcal{N}(v')}{p(x)p_\mathcal{N}(v)}\right)$ is performed to correct for error in the integration process. We have $\frac{q(x, v|x', v')}{q(x', v'|x, v)} = 1$ since the Hamiltonian transition is *volume-preserving* over $(x, v)$. Both HMC steps leave the joint distribution $p(x, v)$ invariant, therefore HMC samples from the correct distribution $p(x)$ after marginalizing over $v$. To express basic HMC in the standard M-H scheme, step 1 and 2 can be aggregated into a single proposal distribution on $x$ with the proposal probability: $q(x'|x) = p_\mathcal{N}(v)$ and $q(x|x') = p_\mathcal{N}(v')$. Note, although the probability $q(x'|x)$ can be calculated after the Hamiltonian dynamics is simulated, this term is intractable for general $x$ and $x'$. The reason is that it is difficult to solve for the $v$ at $x$ to make the transition to $x'$ using the Hamiltonian dynamics. This issue is absent in RWM and MALA, where $q(x'|x)$ is tractable for any $x$ and $x'$.

Previous work on augmenting MCMC sampler with neural networks also relied on the M-H procedure to ensure asymptotic correctness of the sampling process, for example (Song et al., 2017) and (Levy et al., 2018). They used HMC style accept-reject probabilities that lead to intractable $q(x'|x)$. Here, we strive for a flexible sampler for which $q(x'|x)$ is tractable. This maintains the tractable M-H step while allowing us to train this sampler to explores the state space by directly optimizing the proposal entropy objective, which is a function of $q(x'|x)$.

## 3 GRADIENT BASED SAMPLER WITH TRACTABLE PROPOSAL PROBABILITY

We "abuse" the power of neural networks to design a sampler that is flexible and has tractable proposal probability $q(x'|x)$ between any two points. However, without some extra help of the gradient of the target distribution, the sampler would be modeling a conditional distribution $q(x'|x)$ with brute force, which might be possible but requires a large model capacity. Thus, our method uses the gradient of the target distribution. We use an architecture similar to L2HMC (Levy et al., 2018), which itself was inspired by the HMC algorithm and RealNVP Dinh et al. (2016). To quantify

---

[1]Up to ergodic and aperiodic assumptions

the benefit of using the target distribution gradient, we provide ablation studies of our model in the Appendix A.1.

## 3.1 MODEL SETUP AND HOW TO USE GRADIENT INFORMATION

We restrict our sampler to the simple general form $x' = x + z$. As discussed in Section 2, the sampler will have tractable proposal probability if one can calculate the probability of any given $z$. To fulfill this requirement, we model vector $z$ by a flow model[2]: $z = f(z_0; x, U)$, with inverse $z_0 = f^{-1}(z; x, U)$. Here $z_0$ is sampled from a fixed Gaussian base distribution. The flow model $f$ is a flexible and trainable invertible function of $z$ conditioned on $x, U$, and it has tractable Jacobian determinant w.r.t. $z$. The flow model $f$ can be viewed as a change of variable from the Gaussian base distribution $z0$ to $z$. The proposed sampler then has tractable forward and reverse proposal probability: $q(x'|x) = p_Z(x' - x; x), q(x|x') = p_Z(x - x'; x')$, where $p_Z(z; x) = p_{\mathcal{N}}(z_0)|\frac{\partial z}{\partial z_0}|^{-1}$ is the density defined by the flow model $f$. Note, this sampler is ergodic and aperiodic, since $q(x'|x) \neq 0$ for and $x$ and $x'$, which follows from the invertibility of $f$. Thus, combined with the M-H step, the sampling process will be asymptotically correct. The sampling process first consists of drawing from $p_{\mathcal{N}}(z_0)$ and then evaluating $z = f(z_0; x, U)$ and $q(x'|x)$. Next, the reverse $z'_0 = f^{-1}(-z; x + z, U)$ is evaluated at $x' = x + z$ to obtain the reverse proposal probability $q(x|x')$. Finally, the sample is accepted with the standard M-H rule.

For the flow model $f$, we use an architecture similar to a non-volume preserving coupling-based flow RealNVP (Dinh et al., 2016). In the coupling flow, half of the components of the state vector are kept fixed and are used to update the other half through an affine transform parameterized by a neural network. The gradient of the target distribution enters our model in those affine transformations. To motivate the particular form we choose, we take a closer look at the HMC algorithm. Basic HMC starts with drawing a random initial momentum $v^0$, followed by several steps of leapfrog integration. Let $x^n$ be the momentum variable after $n$ updates by $v^n$ and position. In a leapfrog step, the integrator first updates $v$ with a half step of the gradient: $v^{n'} = v^{n-1} - \frac{\epsilon}{2}\partial_x U(x^{n-1})$, followed by a full step of $x$ update: $x^n = x^{n-1} + \epsilon v^{n'}$, and another half step of $v$ update: $v^n = v^{n'} - \frac{\epsilon}{2}\partial_x U(x^n)$. After several steps, the overall update of $x$ can be written as: $x^n = x^0 + \sum_{i=0}^{n} v^{i'}$, which has the form $x' = x + z$ with $z = \sum_i^n v^{i'} = -nv^0 - \frac{n\epsilon}{2}\left[\partial_x U(x^0)\right] - \epsilon\left[\sum_{i=1}^{n}(n-i)\partial_x U(x^i)\right]$. The equation for generating $z$ through affine transformations, describes how the gradient of the target distribution, evaluated at some intermediate point of $x$, should be included.

## 3.2 MODEL FORMULATION

To formulate our model (Equation 1, 2), we use a mask $m$ and its complement $\overline{m}$ to update half of $z$'s dimensions at a time. As discussed above, we include the gradient term with a negative sign in the shift term. We also use an element-wise scaling on the gradient term as in (Levy et al., 2018). However, two issues remain. First, as required by the coupling-based architecture, the gradient term can only depend on the masked version of vector $z$. Second, it is unclear where the gradient should be evaluated to sample effectively. As discussed above, the sampler should evaluate the gradient at points far away from $x$, similar as in HMC, to travel long distances in the state space. To handle these issues, we use another neural network $R$ which depends takes $x$ and the masked $z$ as input, and evaluate gradient at $x + R$. During training, $R$ learns where the gradient should be evaluated based on the masked $z$.

We denote the input to network $R$ by $\zeta_m^n = (x, m \odot z^n)$ and the input to the other networks by $\xi_m^n = (x, m \odot z^n, \partial U(x + R(\zeta_m^n)))$, where $\odot$ is the Hadamard product (element wise multiply). Further, we denote the neural network outputs that parameterize the affine transform by $S(\xi_m^n)$, $Q(\xi_m^n)$ and $T(\xi_m^n)$. For notational clarity we omit dependencies of the mask $m$ and all neural network terms on the step number $n$.

Additionally, we introduce a scale parameter $\epsilon$, which modifies the $x$ update to $x' = x + \epsilon * z$. We also define $\epsilon' = \epsilon/(2N)$, with $N$ the total number of $z$ update steps. This parameterization makes our sampler equivalent to the MALA algorithm with step size $\epsilon$ at initialization, where the neural

---

[2]For more details on flow models, see (Kobyzev et al., 2019; Papamakarios et al., 2019).

network outputs are zero. The resulting update rule is:

$$z^{n\prime} = m \odot z^{n-1} + \overline{m} \odot \left(z^{n-1} \odot \exp[S(\xi_m^{n-1})] - \epsilon'\{\partial U[x + R(\zeta_m^{n-1})] \odot \exp[Q(\xi_m^{n-1})] + T(\xi_m^{n-1})\}\right) \quad (1)$$

$$z^n = \overline{m} \odot z^{n\prime} + m \odot \left(z^{n\prime} \odot \exp[S(\xi_{\overline{m}}^{n\prime})] - \epsilon'\{\partial U\left[x + R(\zeta_{\overline{m}}^{n\prime})\right] \odot \exp[Q(\xi_{\overline{m}}^{n\prime})] + T(\xi_{\overline{m}}^{n\prime})\}\right) \quad (2)$$

The log determinant of N steps of transformation is:

$$\log\left|\frac{\partial z^N}{\partial z^0}\right| = \epsilon\, \mathbf{1} * \mathbf{1} + \sum_{n=1}^{N} \mathbf{1} * \left[\overline{m} \odot S(\xi_m^{n-1})\right] + \mathbf{1} * \left[m \odot S(\xi_{\overline{m}}^{n\prime})\right] \quad (3)$$

where $\mathbf{1}$ is the vector of 1-entries with the same dimension as $z$.

### 3.3 OPTIMIZING THE PROPOSAL ENTROPY OBJECTIVE

The proposal entropy can be expressed as:

$$H\left(X'|X=x\right) = -\int dx' q\left(x'|x\right) \log\left[q\left(x'|x\right)\right] = -\int dz^0 p_{\mathcal{N}}\left(z^0\right)\left[\log\left(p_{\mathcal{N}}\left(z^0\right)\right) - \log\left|\frac{\partial z^N}{\partial z^0}\right|\right] \quad (4)$$

For each $x$, we aim to optimize $S(x) = \exp\left[\beta H(X'|X=x)\right] \times a(x)$, where $a(x) = \int A(x', x) q(x'|x) dx'$ is the average accept probability of the proposal distribution at $x$. Following (Titsias & Dellaportas, 2019), we transform this objective into log space and use Jensen's inequality to obtain a lower bound:

$$\log S(x) = \log \int A(x', x) q(x'|x) dx' + \beta H(X'|X=x)$$

$$\geq \int \log\left[A(x'x)\right] q(x'|x) dx' + \beta H(X'|X=x) = L(x)$$

The distribution $q(x'|x)$ is reparameterizable, therefore the expectation over $q(x'|x)$ can be expressed as expectation over $p_{\mathcal{N}}(z_0)$. Expanding the lower bound $L(x)$ and ignoring the entropy of the base distribution $p_{\mathcal{N}}(z_0)$, we arrive at:

$$L(x) = \int dz^0 p_{\mathcal{N}}(z^0)\left[\min\left(0, \log\frac{p(x')}{p(x)} + \log\frac{q(x|x')}{q(x'|x)}\right) - \beta \log\left|\frac{\partial z^N}{\partial z^0}\right|\right] \quad (5)$$

During training we maximize $L(x)$ with $x$ sampled from the target distribution $p(x)$ if it is available, or with $x$ obtained from the bootstrapping process (Song et al., 2017) which maintains a buffer of samples and updates them continuously. Typically, only one sample of $z^0$ is used for each $x$.

A curious feature of our model is that during training one has to back-propagate over the gradient of the target distribution multiple times to optimize $R$. In (Titsias & Dellaportas, 2019) the authors avoid multiple back-propagation by stopping the derivative calculation at the density gradient term. In our experiment we do not use this trick and perform full back-propagation without encountering any issue. We found that stopping the derivative computation instead harms performance.

The entropy-based exploration objective contains a parameter $\beta$ that controls the balance between acceptance rate and proposal entropy. As in (Titsias & Dellaportas, 2019), We use a simple adaptive scheme to adjust $\beta$ to maintain a constant accept rate close to a target accept rate. The target accept rate is chosen empirically. As expected, we find that the target accept rate needs to be lower for more complicated distributions.

## 4 RELATED WORKS: NEURAL NETWORK MCMC SAMPLERS INSPIRED BY HMC

Here we discuss other neural network MCMC samplers and how they differ from our method. Methods we compare ours to in the Results are marked with **bold font**.

**A-NICE-MC** (Song et al., 2017), which was generalized in (Spanbauer et al., 2020), used the same accept probability as HMC, but replaced the Hamiltonian dynamics by a flexible volume-preserving flow (Dinh et al., 2014). A-NICE-MC matches samples from $q(x'|x)$ directly to samples from $p(x)$,

using adversarial loss. This permits training the sampler on empirical distributions, i.e., in cases where only samples but not the density function is available. The problem with this method is that samples from the resulting sampler can be highly correlated because the adversarial objective only optimizes for the quality of the proposed sample. If the sampler produces a high quality sample $x$, the learning objective does not encourage the next sample $x'$ to be substantially different from $x$. The authors used a pairwise discriminator that empirically mitigated this issue but the benefit in exploration speed is limited.

Another related sampling approach is **Neural Transport MCMC** (Marzouk et al., 2016; Hoffman et al., 2019; Nijkamp et al., 2020) , which fits a distribution defined by a flow model $p_g(x)$ to the target distribution using $\mathbf{KL}[p_g(x)||p(x)]$. Sampling is then performed with HMC in the latent space of the flow model. Due to the invariance of the KL-divergence with respect to a change of variables, the "transported distribution" in $z$ space $p_{g^{-1}}(z)$ will be fitted to resemble the Gaussian prior $p_{\mathcal{N}}(z)$. Samples of $x$ can then be obtained by passing $z$ through the transport map. Neural transport MCMC improves sampling efficiency compared to sampling in the original space because a distribution closer to a Gaussian is easier to sample. However, the sampling cost is not a monotonic function of the KL-divergence used to optimize the transport map. (Langmore et al., 2019).

Another line of work connects the MCMC method to Variational Inference (Salimans et al., 2015; Zhang et al., 2018). Simply put, they improve the variational approximation by running several steps of MCMC transitions initialized from a variational distribution. The MCMC steps are optimized by minimizing the KL-divergence between the resulting distribution and the true posterior. This amounts to optimizing a "burn in" process in MCMC. In our setup however, the exact sampling is guaranteed by the M-H process, thus the KL divergence loss is no longer applicable. Like in variational inference, the **Normalizing flow Langevin MC** (NFLMC) (Gu et al., 2019) also used a KL divergence loss. Strictly speaking, this model is a normalizing flow but not a MCMC method. We compare our method to it, because the model architecture, like ours, uses the gradient of the target distribution.

Another related technique is (Neklyudov et al., 2018), where the authors trained an independent M-H sampler by minimizing $\mathbf{KL}\left[p(x)q(x'|x)||p(x')q(x|x')\right]$. This objective can be viewed as a lower bound of the M-H accept rate. However, as discussed in (Titsias & Dellaportas, 2019), this type of objective is not applicable for samplers that condition on the previous state.

All the mentioned techniques have in common that their objective does not encourage exploration speed. In contrast, **L2HMC** (Levy et al., 2018; Thin et al., 2020) does encourage fast exploration of the state space by employing a variant of the expected square jump objective (Pasarica & Gelman, 2010): $L(x) = \int dx' q(x'|x)A(x',x)||x' - x||^2$. This objective provides a learning signal even when $x$ is drawn from the exact target distribution $p(x)$. L2HMC generalized the Hamiltonian dynamics with a flexible non-volume-preserving transformation (Dinh et al., 2016). The architecture of L2HMC is very flexible and uses gradient of target distribution. However, the L2 expected jump objective in L2HMC improves exploration speed only in well-structured distributions (see Figure 1).

The shortcomings of the discussed methods led us to consider the use of an entropy-based objective. However, L2HMC does not have tractable proposal probability $p(x'|x)$, preventing the direct application of the entropy-based objective. In principle, the proposal entropy objective could be optimized for the L2HMC sampler with variational inference (Poole et al., 2019; Song & Ermon, 2019), but our preliminary experiments using this idea were not promising. Therefore, we designed our sampler that possess tractable proposal probability and investigated tractable optimization of the proposal entropy objective.

## 5 EXPERIMENTAL RESULT

### 5.1 SYNTHETIC DATASET AND BAYESIAN LOGISTIC REGRESSION

First we demonstrate that our technique accelerates sampling of the funnel distribution, a particularly pathological example from (Neal, 2003). We then compare our model with A-NICE-MC (Song et al., 2017), L2HMC (Levy et al., 2018), Normalizing flow Langevin MC (NFLMC) (Gu et al., 2019) as well as NeuTra (Hoffman et al., 2019) on several other synthetic datasets and a Bayesian

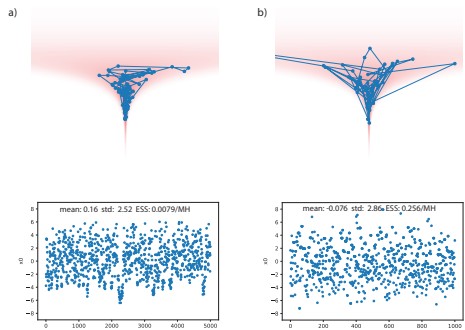

| Dataset (measure) | L2HMC | Ours |
|---|---|---|
| 50d ICG (ESS/MH) | 0.783 | **0.86** |
| 2d SCG (ESS/MH) | 0.497 | **0.89** |
| 50d ICG (ESS/grad) | $7.83 \times 10^{-2}$ | $\mathbf{2.15 \times 10^{-1}}$ |
| 2d SCG (ESS/grad) | $2.32 \times 10^{-2}$ | $\mathbf{2.2 \times 10^{-1}}$ |

| Dataset (measure) | Neutra | Ours |
|---|---|---|
| Funnel-1 $x_0$ (ESS/grad) | $8.9 \times 10^{-3}$ | $\mathbf{3.7 \times 10^{-2}}$ |
| Funnel-1 $x_1 \dots 99$ (ESS/grad) | $4.9 \times 10^{-2}$ | $\mathbf{7.2 \times 10^{-2}}$ |

| Dataset (measure) | gradMALA | A-NICE-MC | NFLMC | Ours |
|---|---|---|---|---|
| German (ESS/5k) | 702.05 | 926.49 | 1176.8 | **3150** |
| Australian (ESS/5k) | 871.5 | 1015.75 | 1586.4 | **2950** |
| Heart (ESS/5k) | 973.2 | 1251.16 | 2000 | **3600** |

Figure 2: Comparison with HMC on the 20d Funnel-3 distribution. a) Chain and samples of $x_0$ (from neck to base direction) for HMC. b) Same as a) but for our learned sampler. Note, samples in a) look significantly more correlated than those in b), although they are plotted over a longer time scale.

Table 1: Performance Comparisons. SCG: strongly correlated Gaussian, ICG: Ill-conditioned Gaussian. German, Autralian, Heart: Datasets for Bayesian Logistic regression. ESS: Effective Sample Size (a correlation measure)

logistic regression task. We additionally compare to gradMALA (Titsias & Dellaportas, 2019) to show the benefit of using neural network over linear adaptive sampler. For all experiments, we report Effective Sample Size (Hoffman & Gelman, 2014) per M-H step (ESS/MH) and/or ESS per target density gradient evaluation (ESS/grad). All results are given in *minimum* ESS over all dimensions unless otherwise noted.

Here is a brief description of the datasets used in our experiments:

**Ill Conditioned Gaussian**: 50d ill-conditioned Gaussian task described in (Levy et al., 2018), a Gaussian with diagonal covariance matrix with log-linearly distributed entries between $[10^{-2}, 10^2]$.

**Strongly correlated Gaussian**: 2d Gaussian with variance $[10^2, 10^{-1}]$ rotated by $\frac{\pi}{4}$, same as in (Levy et al., 2018).

**Funnel distribution**: The density function is $p_{funnel}(x) = \mathcal{N}(x_0; 0, \sigma^2)\mathcal{N}(x_{1:n}; 0, \mathbf{I} \exp(-2x_0))$. This is a challenging distribution because the spatial scale of $x_{1:n}$ varies drastically depending on the value of $x_0$. This geometry causes problems to adaptation algorithms that rely on a spatial scale. An important detail is that earlier work, such as (Betancourt, 2013) used $\sigma = 3$, while some recent works used $\sigma = 1$. We run experiments with $\sigma = 1$ for comparison with recent techniques and also demonstrate our method on a 20 dimensional funnel distribution with $\sigma = 3$. We denote the two variants by Funnel-1 versus Funnel-3.

**Bayesian Logistic regression**: We follow the setup in (Hoffman & Gelman, 2014) and used German, Heart and Australian datasets from the UCI data registry.

In Figure 2, we compare our method with HMC on the 20d Funnel-3 distribution. As discussed in Betancourt (2013), the stepsize of HMC needs to be manually tuned down to allow traveling into the neck of the funnel, otherwise the sampling process will be biased. We thus tune the stepsize of HMC to be the largest that still allows traveling into the neck. Each HMC proposal is set to use the same number of gradient steps as each proposal of the trained sampler. As can be seen, the samples proposed by our method travel significantly further than the HMC samples. Our method achieves 0.256 (ESS/MH), compared to 0.0079 (ESS/MH) with HMC.

As a demonstration we provide a visualization of the resulting chain of samples in Figure 2 and the learned proposal distributions in Appendix A.2. The energy value for the neck of the funnel can be very different than for the base, which makes it hard for methods such as HMC to mix between them (Betancourt, 2013). In contrast, our model can produce very asymmetric $q(x'|x)$ and $q(x|x')$, making mixing between different energy levels possible.

Performances on other synthetic datasets and the Bayesian Logistic Regression are shown in Table 1. In all these datasets our method outperformed previous neural network based MCMC approaches by significant margin. Our model also outperform gradMALA (Titsias & Dellaportas, 2019), which

a)
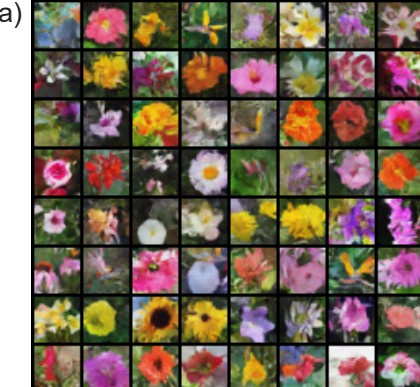

b)
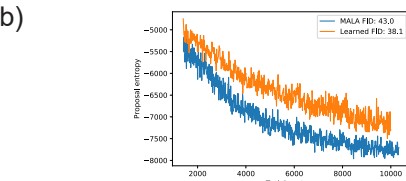

c)
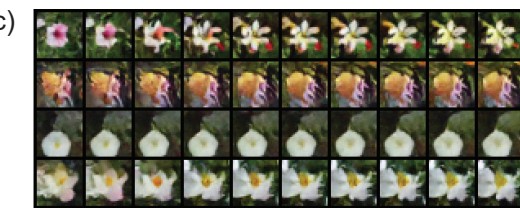

Figure 3: Training of convergent EBM with pixel space sampling. a) Samples from replay buffer after training. b) Proposal entropy of trained sampler vs MALA early during training, learned sampler has significantly higher entropy, and achieves better FID at convergence. c) Samples from 100k sampling steps by the learned sampler, initialized at samples from replay buffer. Large transitions like the one in the first row is rare, here its selected for display.

use the same objective but only use linear adaptive parameters. The experiments used various parameter settings, as detailed in Appendix A.2. Results of other models are adopted or converted from numbers reported in the original papers. The Appendix provides further experimental results, ablation studies, visualizations and details on the implementation of the model.

## 5.2 TRAINING A CONVERGENT DEEP ENERGY-BASED MODEL

A very challenging application of the MCMC method is training a deep energy-based model of images (Xie et al., 2016; Nijkamp et al., 2019; Du & Mordatch, 2019). We demonstrate stable training of a convergent EBM, and that the learned sampler achieves better proposal entropy early during training, as well as better sample quality at convergence, compared to the MALA algorithm. An added benefit is that, like in adaptive MALA, tuning the Langevin dynamics step size is no longer needed, instead, one only need to specify a target accept rate. This contrast with unadjusted Langevin dynamics used in previous works, where step size need to be carefully tuned(Nijkamp et al., 2019).

Similar to (Nijkamp et al., 2019), we use the Oxford flowers dataset of 8189 $28*28$ colored images. We dequantize the images to 5bits by adding uniform noise and use logit transform (Dinh et al., 2016). Sampling is performed in the logit space with variant 2 of the sampler that does not have $R$ network (See Appendix A.1). During training, we use Persistent Contrastive Divergence (PCD) (Tieleman, 2008) with replay buffer size of 10000. We alternate between training the sampler and updating samples for the EBM training. Each EBM training step uses 40 sampling steps, with a target accept rate of 0.6.

Figure 3 depicts samples from the trained EBM replay buffer, as well as samples from a 100k step sampling process –for demonstrating stability of the attractor basins. We also show that the proposal entropy of the learned sampler is higher early during training than that of an adaptive MALA algorithm with the same accept rate target. Later during training, the proposal entropy is not significant different (See Figure A.3 a)). This is likely because the explorable volume around samples becomes too small for the learned sampler to make a difference. Additionally, we show the model trained with the learned sampler achieves better sample quality by calculating the FID (Heusel et al., 2017) between the replay buffer for a late checkpoint and ground truth data. Model trained with learned sampler achieves 38.1 FID, while model trained with MALA achieves 43.0 FID (lower is better). We provide a plot that tracks the FID during training in Appendix Figure A.3.

## 6 DISCUSSION

In this paper we propose a gradient based neural network MCMC sampler with tractable proposal probability. The training is based on the entropy-based exploration speed objective. Thanks to an objective that explicitly encouraging exploration, our method achieves better performance than previous neural network based MCMC samplers on a variety of tasks. Compared to the manifold HMC (Betancourt, 2013) methods, our model provides a more scalable alternative for mitigating unfavorable geometry in the target distribution.

There are many potential applications of our method beyond what was demonstrated in this paper. For example, training latent-variable models (Hoffman, 2017), latent sampling in GANs (Che et al., 2020), and others applications outside machine learning, such as molecular dynamics simulation (Noé et al., 2019). In the future, architectural improvement would also be interesting, use of auto-regressive architecture or different masking strategy may improve the expressiveness of our model. It will also be interesting to combine our technique with neural transport MCMC.

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
