# OpenReview forum: "A Neural Network MCMC sampler that maximizes Proposal Entropy"
_ICLR.cc/2021/Conference — Reject_

### Official Review · AnonReviewer4 · 2020-10-27
**Interesting paper**

**Rating:** 6
**Confidence:** 3

**Review:**

The paper argues that a better objective to train neural MCMC kernels is to maximize the proposal entropy (Titsias & Dellaportas, 2019) and demonstrate a method on doing so. The method shows improved sampling efficiency compared to previous method, especially one that optimize the alternative L2 expected jump. The novelty is not of the training objective but a neural instantiation with improved sampling efficiency.

## Pros

1. The method is well-motivated in Section 1 and clearly demonstrated by Figure 1.
2. The neural instantiation consists a few clever tricks to make the network tractable and explore the space well.
3. The method demonstrates improved sampling efficiency, measured by ESS.

## Cons

1. The paper may need more work on presentation.
- Section 3 is hard to follow. It's very long and maybe adding some subsection would help.
- I feel Section 4 should go before Section 3, given how it is currently presented.

2. There is a false statements in the paper: "one can easily make any transition probability satisfy it by including an additional Metropolis-Hastings accept-reject step (Hastings, 1970)"
- It's not true. One need to additionally ensure that the kernel is irreducible and aperiodic.
- For the reason above, it's worth the mention that the proposed neural approach is irreducible and aperiodic, and give some justificaiton.

3. No experiments for correctness check. I understand that showing better ESS is good, but even some biased sampler can lead to much improved ESS. I think there should be some results on comparing the proposed method against well-established HMC on a few Bayesian inference problem in terms of the posterior. Using some hypothesis testing methods to check if the posterior actually matches would give me more confidence that the method is sampling from the target; or at least, comparing a few moments.

## Questions

1. "In (Titsias & Dellaportas, 2019), the authors avoid multiple back-propagation by stopping the derivative calculation at the density gradient term. In our experiment, we find it is necessary for good performance.": does it mean you effectively using a biased gradient and it turns out to be better than an unbiased one? It looks quite weird to me as it means you are in fact optimizing some other objective which turns out to be better.

2. I'm generally unsatisfied with the setup of Section 5.2.
- "which removes the need to choose the step size. The only tunable parameter is a target accept rate.": I think vanilla HMC or MALA can do so by some online otpimization for step size as well (e.g. dual averaging as it's done in NUTS).
- The unstable training of EBMs via contrastive divergence comes from the fact that the gradient estimated by a short-run MCMC is **unbiased**. The proposed method itself doesn't deal with the biasness directly, so the only plausible reason to explain why it improves the training is that the mixing is so good such that with the short chain, the bias of the gradient is so small. Is this what happening?
- I wonder whether or not the simultaneous/interweaved training of samplers and EBMs would cause any issue. In particular, will it encourage that EBM only putting energy around data points and the sampler only effectively draw samples from some noisy distribution nearby? Are samples from EBMs too similar to data points.
- For Figure 3.b, what's the convergence behaviour, i.e. do learnable sampler and MALA converge to a similar entropy or not?
- For Figure 3.c, does long-run MALA produce sensible samples? If yes, it indicates that this way of training EBMs avoid some pathology which CD with short-run MCMC has; if no, then it may indicate that somehow the learnable sampler co-adapt with the EBM somehow to avoid the pathology. Is the latter we wanted?

---

> ### Author Response · Authors · 2020-11-20
> **Response to Reviewer 4**
>
> Thanks for your detailed review and suggestions. Here our responses:
>
> 1. Presentation of section 3 and 4
>
> Thanks for the suggestion! We divided section 3 into subsections and made several improvements on the presentation of our model. We also moved the discussion of HMC in section 4 to section 2, which logically makes much more sense. We still keep the discussion of other neural network based models after section 3, which is easier to digest after the introduction of our own model. Hopefully, our manuscript is now much easier to follow and you are satisfied with the new structure.
>
> 2. Discuss irreducibility and aperiodicity of kernel
>
> We thank the reviewer for pointing this out. We have updated our draft to address this issue. It is straightforward to see that our proposal distribution parameterized by the flow model is ergodic and aperiodic. As the probability of reaching any point x1 from any other point x0 is non-zero, this follows easily by construction as the flow model is invertible. See also Section 3 of our updated manuscript.
>
> 3. Correctness Check of samples
>
> We have added the experiment that compares dimension-wise mean, std and 4th moment of samples generated by the learned sampler and samples from HMC. The result matches very well, indicating that our method samples from the correct target distribution. See Appendix Figure 4 a).
>
> 4. Multiple back-propagation
>
> We apologize for making an ambiguous statement in our paper. By “It is important for good performance”, we are actually referring to back-propagating through all the gradient computation, instead of using the stop-gradient trick. Just to clarify, we actually performed full back-propagation through multiple gradient computations, and this is important for achieving good performance. We have fixed this wording issue in our updated draft.
>
> 5. Compare to adaptive HMC and MALA
>
> We agree that HMC and MALA can also adapt step size, making the accept rate the only tunable parameter. Thus, not needing to choose a step size shouldn’t be described as  an advantage of our model, we have adjusted the wording accordingly in the revised manuscript.
>
> 6. Stability of EBM training
>
> In further experiments with MALA we realized that in our particular setup MALA can also stably train the model to convergence. Therefore, the main advantage of our method is not saving sampling steps but rather achieving higher proposal entropy and better sample quality. Please see the general response about updates on the paper as well as the updated draft.
>
> 7. Simultaneous training of sampler and EBM, MALA long run samples
>
> We believe that interleaved training of the sampler and EBM do not cause problems, because the sampler is trained to explore the state space more thoroughly. It is true that the sampler is only exploring a small area near the current data point, but the same is true for the MALA sampler - and there is yet to be a sampler that can travel between modes effectively for image EBM training.
> The goal of maximum likelihood training is to only put low energy on plausible samples, and increase the energy of everything else. The latter is achieved by increasing the energy of samples from the current energy function. In this light, having better mode exploration will help achieve this goal and help training.
> As you mentioned in the last question, it is possible that the learned sampler explores the state space in a particular way such that although the entropy is higher, some area that should be explored is not explored by the learned sampler. However, we verified that this is not the case by sampling an EBM energy function trained by the learned sampler by the MALA algorithm. We find that the resulting samples are still plausible, see Figure A.4 in the appendix.

---

> > ### Comment · AnonReviewer4 · 2020-11-23
> > **Thanks for the response**
> >
> > Thanks for the confirmations regarding multiple back-propagation and stability of EBM training.
> >
> > Re. Simultaneous training of sampler and EBM, MALA long run samples
> >
> > "We believe that interleaved training of the sampler and EBM do not cause problems, because the sampler is trained to explore the state space more thoroughly. It is true that the sampler is only exploring a small area near the current data point, but the same is true for the MALA sampler - and there is yet to be a sampler that can travel between modes effectively for image EBM training."
> >
> > My point is that for MALA, there is nothing to learn for the sampler, so there is no such a concern. More precisely, my question is: for the optimisation problem of training a sampler with parameter $\theta$ and an EBM with parameter $\phi$ together, is the optimiser $\phi^\ast$ same as that of the optimisation problem of training only an EBM with parameter $\phi$ using sufficiently long MALA? I don't think it's easy to answer from a principle point of view, but I do agree with your reasoning and am happy with the empirical results you added to support it.
> >
> > **Additional comments**
> >
> > - After the changes, it seems that the main benefit of the proposed method is "higher proposal entropy and better sample quality". Is my understanding correct?
> > - How does the neural sampler generalise, i.e. training on one target and test on another?

---

> > > ### Author Response · Authors · 2020-11-23
> > > **Response to additional comments**
> > >
> > > Thanks for your response.
> > >
> > > To your first question, whether having a trainable sampler changes the optimal EBM. In our opinion, the answer is false, that is, the optimal EBM does not depend on the sampler. The reason is, the construction of the M-H sampler guarantees the same equilibrium distribution regardless of the parameters of the sampler. If a sampler is run enough steps, they reach the same distribution as MALA.
> > >
> > > In practice, ineffective mixing of the sampler will bias EBM training. However, as our learned sampler is initialized as MALA, and achieves better proposal entropy, it is unlikely to be more biased than MALA due to ineffective mixing. We believe the improved final sample quality for the EBM comes from better mixing allowed by the sampler, which allows more effective "unlearning" of unwanted modes. Hope you do not find it contradicting the point above.
> > >
> > > "After the changes, it seems that the main benefit of the proposed method is "higher proposal entropy and better sample quality". Is my understanding correct?"
> > >
> > > Yes, for EBM experiment this summarizes the benefit. The argument for sampling step saving is no longer valid in light of new experiment, therefore we removed it.
> > >
> > > "How does the neural sampler generalise, i.e. training on one target and test on another?"
> > >
> > > We do not think it generalizes, unless in very special cases. Because the key to the performance of the learned sampler is the dependence of the proposal distribution p(x'|x) on the current location x. This allows the sampler to adapt to different local geometry of the target distribution. As the local geometry is different for different distributions, it is unlikely that a sampler trained on one distribution will be efficient on a different distribution.

---

### Official Review · AnonReviewer3 · 2020-10-28
**Interesting MCMC samplers but need some improvements in empirical analysis.**

**Rating:** 6
**Confidence:** 3

**Review:**

Summary:
The author proposes a novel MCMC sampler parametrized by the neural networks. In particular, the neural network is chosen to be a flow-based model that allows the exact evaluation of the proposal probability.

The update equations of MCMC mimic the dynamics of HMC by incorporating the gradient of the energy function into the flow. To ensure the invertibility, each update equation only depends on the other part of the variables. In addition, the author also designs another network $R$ to propose an evaluation location for the target gradient. This ensures the similarity between the proposed method and HMC.

As for the training objective, the author proposes to maximize the proposal entropy and acceptance rate, controlled by coefficient $\beta$. Due to the tractability of the proposal density, this objective can be analytically computed. Stop gradient trick is used to stabilize the training and reduce the cost of back-propagation.

Empirically, the author evaluates the proposed sampler in some toy datasets, logistic regression, and deep energy-based models. The proposed sampler achieves higher ESS compared to baselines.

---------
Review:
Clarity: The paper is clearly written and easy to follow. The author also addresses the related work and mention the difference compared to previous baselines.

Technical soundness:
I have a quick look at the details of the proposed method. It seems the derivation is correct.

Novelty:
Although the structure of the sampler is inspired by L2HMC, there are still some differences. L2HMC generalizes the HMC update equation by partitioning $\pmb{x}$ into two parts, each part is parameterized by neural networks. The proposed sampler instead partitions the state $\pmb{z}$ instead of parameter $\pmb{x}$ with additional network $R$. The idea of using entropy as an objective to encourage exploration is not new.  However, the novelty lies in the usage of the flow model to allows tractability of proposal density. Overall, the proposed method is novel to some extent.

Significance of the work:
The proposed method can be regarded as a variant of L2HMC with slightly different NN parameterization and training objectives. The advantage of the proposed method is the higher ESS compared to previous samplers. This may be helpful to some audiences.

Weakness:
1. Although the author demonstrates the better ESS can be obtained using the proposed method, I still prefer more analysis to understand the properties of this sampler. For example, I am curious to know how important the network $R$ is? If $R$ is removed, and the leapfrog integrator is used (i.e. update $z^{n'}$ followed by updating $x^{n}$ and finally $z^{n}$ to mimic HMC update rule). Then the gradient evaluation is also evaluated at different locations. What are the differences in terms of performances?
2. The reason that the proposed method can have higher ESS is due to the maximization of the proposal entropy. Although the training objective has an acceptance rate term, I am curious to know: does the entropy term hurts the sample quality? For example, in logistic regression, the author only reports the ESS. What about the convergence speed of the sampler compared to others that use sample quality as the training objective? What about their performances in logistic regression?
3. For training deep EBM, apart from ESS. I also want to know the convergence speed of the EBM training and the quality of the generated images compared to MALA. These are the standard evaluation metric for EBM.
4. For EBM, why only compared to MALA. Any reasons why excludes other baselines?

---

> ### Author Response · Authors · 2020-11-20
> **Response to Reviewer 3**
>
> Thanks for your detailed review. We would like to respond to the following statements.
>
> 1. The stop gradient trick is used to stabilize the training and reduce the cost of back-propagation.
>
> We apologize for making an ambiguous statement in our paper. By “It is important for good performance”, we are actually referring to back-propagating through all the gradient computation, instead of using the stop-gradient trick. Just to clarify, we actually performed full back-propagation through multiple gradient computations, and this is important for achieving good performance. We have fixed this wording issue in the updated draft.
>
> 2. Role of network R and compare with HMC baseline.
>
> Our method takes inspiration from HMC and L2HMC, but importantly, HMC-style update does not admit tractable proposal probability computation, thus the proposal entropy objective cannot be optimized directly.
> In our model, we only update z when generating z but not x. If we would update x depending on z (as if z is the momentum used in HMC), and employ the new x to update z again, the overall probability would become intractable. It is precisely this difficulty that motivated us to redesign our architecture compared to L2HMC.
> In our new architecture, only half of z’s dimension can be updated in each step, using a function that only depends on the other half of the dimension of z. Therefore, in some sense, the neural network R “fills in” the other half of the vector. The only alternative would perhaps be to use the masked z directly and evaluate the gradient at x+z. This would push half of x’s components far away while not changing the other half,  therefore we do not expect this to work well.
>
> 3. Does the entropy term hurt sample quality
>
> When the proposal entropy is high, there will be a mismatch between proposal distribution and the target distribution, however, the M-H accept-reject step corrects this bias, so asymptotically the samples are exactly correct, just like in other MCMC methods.
>
> 4. Convergence speed of sampler in Logistic regression
>
> Unlike L2HMC, we do not optimize a “burn in” process, as we found it can cause numerical instability if the initial distribution is too far from the target distribution. It is therefore not possible to measure the convergence speed.
>
> 5. Convergence speed and sample quality of EBM experiment
>
> Thanks for your suggestion! We measured the sample quality of a model trained with the  learned sampler and that trained with MALA. Indeed, the learned sampler improves sample quality of the final model. See our general comment on paper update as well as the updated paper.
> The convergence speed, however, is harder to measure. There is no generally accepted criteria for convergence of generative models other than sample quality. GANs for example, only use sample quality to measure convergence. Therefore, the improvement on sample quality in some sense reflects a faster convergence, but unfortunately we could not come up with a direct way to measure convergence speed.
>
> 6. Compare to other sampling methods for EBM
>
> Previous works on image EBMs mostly used unadjusted Langevin dynamics [Xie 2016, Du 2019, Nijkamp 2019]. There is no rejection process in unadjusted Langevin dynamics,  therefore it is hard to make a fair comparison. Another alternative method is HMC, but HMC does not have a tractable proposal probability, therefore we cannot compare the proposal entropy directly. Furthermore, it was reported before that it is difficult to apply HMC to image EBMs [Du 2019], so we did not try it in our experiments.
>
> References:
> Erik Nijkamp, Mitch Hill, Tian Han, Song-Chun Zhu, and Ying Nian Wu. On the anatomy of mcmc-based maximum likelihood learning of energy-based models.  InProceedings of the Conferenceon Artificial Intelligence (AAAI), 2019.
>
> Jianwen Xie,  Yang Lu,  Song-Chun Zhu,  and Yingnian Wu.   A theory of generative convnet.   InInternational Conference on Machine Learning, pp. 2635–2644, 2016.
>
> Yilun Du and Igor Mordatch.  Implicit generation and generalization in energy-based models.  InAdvances in Neural Information Processing Systems (NeurIPS), 2019.

---

> > ### Comment · AnonReviewer3 · 2020-11-24
> > **Response to the author**
> >
> > I appreciate the author's response and additional experiments. It indeed addresses most of my concerns. However, I am still curious about the convergence speed. Indeed, there is no clear way to measure this speed, but one can do it by measuring the sample quality at different training times. Also, if the learned sampler does not optimize the burn-in period, will this give a slow burn-in compared to other samples like L2HMC? Anyway, I raised my score to 6.

---

> > > ### Author Response · Authors · 2020-11-25
> > > **Response to reviewer 3**
> > >
> > > Thanks for your reply and increased rating of our paper.
> > >
> > > To answer your first question, we have added the experiment that measures FID throughout training process. The result in shown in updated Figure A3 b). Indeed the learned sampler allows the EBM to converge faster as measured by FID, especially early during training where the difference in proposal entropy is significant.
> > >
> > > To your second question. Since we did not optimize a burn in process, it could give slower burn in than L2HMC, although we did not measure this. However, we do not believe this is a major weakness. Since the bootstrap learning process naturally gives a sample bank of reasonable samples, which can be used to initiate sampling process for more samples.

---

### Official Review · AnonReviewer2 · 2020-10-28
**Interesting paper, but somewhat convoluted description of the algorithm**

**Rating:** 6
**Confidence:** 3

**Review:**

The authors describe an approach for adaptive MCMC which uses a proposal distribution parameterized by a neural network and which optimizes the entropy of the resulting proposal. Overall I found it to be an interesting algorithm, with fairly good results as compared to alternative methods, however the description of the algorithm itself was somewhat confusing and/or convoluted.

The introduction (including section 2) was great, and provided a concise but very readable introduction to recent approaches for adaptive MCMC. The beginning of section 3 was also quite well presented. However, when the authors turn to the parts of their approach inspired by HMC it gets a bit murkier. Part of the difficulty here is that at this point (the first paragraph of p4) the authors are describing HMC rather than their method, which is not entirely clear. Perhaps this could have been simplified by moving the "related work" section earlier in the paper and allowing for the description of HMC before diving into their own approach.

The relation between HMC and their approach which makes use of intermediate steps x+R could also have been more thoroughly explained and given more intuition. The link is there, but it doesn't seem to be as close to the leapfrog step as x+R doesn't correspond to an intermediate step (which would be x+z^n). Not that I'm saying in any way that this makes the algorithm incorrect, just that the connection isn't quite as clear cut.

Similarly I would also have liked to see a more clear description of the Q, T, and S matrices.

Throughout this work the authors also refer to "during training". I assume they perform their adaptation steps during the sample process as is the case of the Titsias and Dellaportas work, however this could use some clarification. Overall, I think that this and other confusions cited above could have been done away with by including a clearer outline/overview of the algorithm as a whole.

Finally, overall the results seem to be quite a bit better than competing methods (although I'm not an expert in this area). However, although the authors hasted to add that they do not compare "exact computation time or ESS/second" it would have been nice to see a more thorough discussion of the relative computational complexity of the alternatives. While ESS/grad does in some sense get close to this, the computation necessary for networks (depending on their size) could play a factor here. Similarly, I would like to see more discussion with regards to the choice of architecture for these networks---ie were they a simple MLP? It's possible I missed this, but do not think it was discussed.

---

> ### Author Response · Authors · 2020-11-20
> **Response to Review 2**
>
> Thanks for your thoughtful comments and suggestions. We would like to make the following responses.
>
> 1. Presentation Issue in Section 3 and 4
>
> Thanks for the suggestion. We have significantly improved section 3 and moved the description of HMC to section 2, which provides better logical flow. We hope you are satisfied with the changes.
>
> 2. Connection to HMC
>
> You are correct that our method only vaguely relates to HMC, and the correspondence is not exact. HMC mostly serves as a motivation in the design of our model. We have updated section 3 to make this clearer.
>
> 3. S, Q, T and network architecture.
>
> Indeed, S, Q, T as well as R used simple MLP architecture, except in EBM training where we used convolutional architecture. We have updated the related section in Appendix to more clearly explain this.
>
> 4. Computation cost comparison with other methods.
>
> It is difficult to compare the computation cost of our method to others without having a working implementation. You are certainly correct that neural network width plays an important role. A-NICE-MC for example, seems to use wider networks: 400 neurons in the Logistic regression tasks, quite a bit more than what we used. Considering that they used more flow steps and achieved lower ESS, their method is likely less efficient. However, it is unclear how the overall cost compares due to the need of evaluating target gradient. L2HMC uses narrower networks (100 neurons in 50d ICG task), but it is only run on small synthetic datasets, making comparison also unsatisfactory.

---

### Official Review · AnonReviewer1 · 2020-10-29

**Rating:** 3
**Confidence:** 2

**Review:**

This paper proposes a new MCMC transition kernel. This kernel is parameterized by neural networks and is optimized through an objective maximizing the proposal entropy. Specifically, the authors use a combination of a flow model and non-volume preserving flow in [Dinh et al., 2016] as the neural network parameterized kernel. Then they use the objective in [Titsias & Dellaportas, 2019] which maximizes the proposal entropy to optimize the kernel. The proposed method is tested on synthetic datasets, Bayesian logistic regression and a deep energy-based model.

The problem of improving the exploration efficiency of MCMC kernel is important. The proposed method is well-motivated. As far as I understand, the proposed method appears to be technically sound.

However, I have the following concerns about the paper.

-	The connection and the difference to previous work are not very clear. If I understand it correctly, the proposed method seems a combination of L2HMC and [Titsias & Dellaportas, 2019] with some slight modification (a flow model) since the naïve combination did not work well (as stated in Section 4). I think it would improve the clarity a lot if the authors explain more clearly how the proposed method differs from previous work.
-	Since the use of a flow model is the main difference compared to the naïve combination of two previous methods, the authors should explain more about this choice. Currently, it is not clear why this helps and there is no explanation on why the naïve combination fails.
-	The proposed method seems to use more neural networks (e.g. an additional network R to handle gradient) than previous neural network MCMC. I wonder how the method performs if considering the cost. For example, the authors may instead show ESS per second on the experiments in Section 5.1. Though the authors mentioned the difficulty of computing ESS per second in the paper, I’m not entirely convinced. As the experiments in Section 5.1 are all very small-scale, is it really necessary to use GPUs?

-	The baselines vary from experiments to experiments for no reason. For example, the authors compare their method to L2HMC on ill-conditioned Gaussian and strongly correlated Gaussian, to Neutra on Funnel distribution, and to MALA on EBM. I think this experiment design needs explanation. Also, there is no empirical comparison to [Titsias & Dellaportas, 2019] which is closely related to the proposed method.

Some minor comments:

-	A.1 intends to show the benefit of using gradient information. But Variant 2 also uses gradient. What is the point of showing it?
-	It is not clear to me how to interpret the empirical results in Section 5.2. For example, how does Figure 3 show that the proposed method needs less sampling steps?
-	The color of points in figure 1 is hard to read.

---

> ### Author Response · Authors · 2020-11-20
> **Response to review1**
>
> We thank the reviewer for evaluation and detailed comments.  Here are our responses to some of the statements and raised concerns:
>
> 1. “the proposed method seems a combination of L2HMC and [Titsias & Dellaportas, 2019]”
>
> It is correct that our method took inspiration from L2HMC, but it features important differences. As explained in the beginning of section 4 (section 2 in updated draft), HMC, and also L2HMC, do not have tractable proposal distributions. Thus, while we kept the core idea of HMC (to evaluate gradient at intermediate sample locations),  we had to completely redesign the architecture to ensure tractability of the proposal distribution.
>
> 2. Explain the use of flow models and “Currently, it is not clear why this helps and there is no explanation on why the naïve combination fails”.
>
> Our model uses a flow model in order to have tractable proposal distribution. L2HMC and A-NICE-MC also use flow models, but their proposal distribution p(x’|x) remains intractable because  the flow models are in the joint space (x,v).
> If we understood correctly, by naive combination you mean to optimize the proposal entropy of L2HMC with variational inference, a method we refer to at the end of section 4. This method, more specifically variational mutual information estimation, has several disadvantages. First, since the proposal entropy needs to be optimized for each sample location x, the “discriminator” required in this method also needs to condition on x. For this to work, a very high capacity neural network would be required. Second, as was discussed in [Song&Ermon 2019], such variational estimators have high variance in the large MI regime, making the training signal very noisy. In our experiment, this approach worked very poorly so we did not explore it further.
>
> 3. Cost of neural networks compared to previous methods and the use of GPUs.
>
> We first note that previous approaches, such as L2HMC and NICE-MC, used GPUs, because evaluating the neural networks can be expensive. Especially during training a lot of evaluations are necessary. This is also the case for our method.
> We did not perform ESS/second comparisons with previous methods because computation time on a GPU depends critically on factors such as batch size, the particular evaluation kernel, etc, making such comparisons somewhat arbitrary. We thus used ESS/MH and ESS/Grad because they do not suffer from this problem. Another reason is, comparing ESS/second requires reproducing the previous models exactly on our setup, which can be tricky.
> We also want to note that using the additional network R does not necessarily make our method more expensive, as the cost of evaluating a neural network also depends on its depth and width.
>
> 4. Different baselines and comparison to [Titsias & Dellaportas, 2019]
>
> To compare with different previous models, we used different baselines because because also the studies describing the models  used distinct baselines. Reproducing previous models the reported performances can be tricky and time consuming. We hope that improving upon previous methods on reported performances is a strong enough indication that our method performs better.
> We agree that we should also compare with [Titsias & Dellaportas, 2019]. We have added the comparison in our paper revision. See updated section 5. Our method clearly outperformed gradMALA from this work, demonstrating the benefit of using neural networks versus simple linear adaptive parameters.
>
> 5. Why compare variant 2
>
> We used variant 2 in the EBM experiment, because it uses less gradient evaluation and does not involve network R, which greatly helps the speed. We explained this point only in the appendix. In response to your comment, we have also explained this in the main text.
>
> 6. How does Figure 3 show that the proposed method needs less sampling steps?
>
> Figure 3, b) shows that our sampler achieves significantly higher proposal entropy than the MALA sampler. After further experiment, we found that the advantage of the learned sampler is not on the number of steps required. The improvement is in terms of proposal entropy early during training and on sample quality. See our general comment on updates on the paper as well as the updated section 5.
>
> References:
>
> Jiaming Song and Stefano Ermon.  Understanding the limitations of variational mutual informationestimators.arXiv preprint arXiv:1910.06222, 2019.
>
> Michalis Titsias and Petros Dellaportas.   Gradient-based adaptive markov chain monte carlo.   InAdvances in Neural Information Processing Systems, pp. 15730–15739, 2019.

---

> > ### Comment · AnonReviewer1 · 2020-11-24
> > **Thank you for your response**
> >
> > Thanks for the explanation. I still have the following concerns.
> >
> > - Based on the response, it seems like you directly used the reported results from previous papers, instead of implementing them yourselves. I think you should explicitly say that in the paper.
> >
> > - I agree that implementing previous methods could be time-consuming, but this will make sure that other factors are the same (e.g. environment, package versions, language, etc.) so that the comparison is fair. The need of implementing previous methods should not be the reason to use different baselines for different experiments. In fact, using the reported results directly from prior work makes the comparison less convincing in some sense, since the improvement could come from many other factors.
> >
> > - Because of the similar reason above, I'm not convinced by the explanation of not comparing ESS/second. I do not think the comparison will be "tricky" if you implement methods correctly. You may also submit code to support your comparison.
> >
> > - Different depth and width in neural network R can definitely affect running time. I'm curious how R affects the algorithm efficiency in the experiments now. I do not find answers for it.

---

> > > ### Author Response · Authors · 2020-11-25
> > > **Response to comments from Reviewer 1**
> > >
> > > "Based on the response, it seems like you directly used the reported results from previous papers, instead of implementing them yourselves. I think you should explicitly say that in the paper"
> > >
> > > In the last paragraph of Section 5.1 we stated :"Results of other models are adopted or converted from numbers reported in the original papers." This statement is also in the original draft.
> > >
> > > "In fact, using the reported results directly from prior work makes the comparison less convincing in some sense, since the improvement could come from many other factors."
> > >
> > > What you said is true if we are comparing ESS/sec, since it’s a measure that depends on the particular computer setup/package versions, etc. For comparing ESS/sec, It would be necessary to reimplement previous models on our setup. ESS/MH and ESS/grad however, is an intrinsic property of the sampler, which directly reflects how efficiently the sampler explores the state space in each proposal and each gradient use, therefore its invariant to factors external to the algorithm like computer setup and package versions. It is precisely because of this that we used the latter two metrics to compare with previous models to avoid reimplementation burden.
> > >
> > > "I do not think the comparison will be "tricky" if you implement methods correctly."
> > >
> > > This could be a misunderstanding. We do not believe the comparison itself is tricky, the tricky part is reimplementing previous models. L2HMC for example, used tensorflow which we do not have expertise in. Besides, in our early experiments, we found L2 jump loss very tricky to use, therefore we are not confident we can tune our reimplemented model to the same performance as reported in the previous papers, and using the original numbers seems more prudent.
> > >
> > > Besides, other recent neural MCMC papers, Neutra [Hoffman 2019] for example, also did not compare ESS/sec with other neural MCMC techniques. Therefore we believe it's not entirely fair to require that from us.
> > >
> > > To provide confidence to you that our method does achieve improvement in terms of computation speed, we added ESS/sec comparison to HMC in the Appendix, please see Table A2.
> > >
> > > "Different depth and width in neural network R can definitely affect running time. I'm curious how R affects the algorithm efficiency in the experiments now. I do not find answers for it."
> > >
> > > We did not try to minimize the size of the R network and simply used the same size as the network for S,Q and T. We did not study how the size of R network would affect performance, a much smaller size could be sufficient, which would save computation time further. However, as shown in Table A2, our simple choice already achieves better performance than HMC.
> > >
> > > Hoffman, Matthew, et al. "Neutra-lizing bad geometry in hamiltonian monte carlo using neural transport." arXiv preprint arXiv:1903.03704 (2019).

---

### Public Comment · ~Jianwen_Xie1 · 2020-11-14
**related works**

Dear Authors and Reviewers,

We found that the current paper missed some important references about pioneering works that are related to energy-based generative models parameterized with deep net energy.

The first paper that proposes to train an energy-based model parameterized by modern deep neural network and learned it by Langevin based MLE is in (Xie. ICML 2016) [1]. The model is called generative ConvNet, because it can be derived from the discriminative ConvNet. This is also the first paper to formulate modern ConvNet-parametrized EBM as exponential tilting of a reference distribution, and connect it to discriminative ConvNet classifier. That is, EBM is a generative version of a discriminator. (Xie. ICML 2016) [1] originally studied such an EBM model on image generation theoretically and practically in 2016.

(Xie. CVPR 2017) [2] (Xie. PAMI 2019) [3] proposed to use Spatial-Temporal ConvNet as the energy function in EBMs for video generation by MCMC. The model is called Spatial-Temporal generative ConvNet.

(Xie. CVPR 2018) [4] also proposed to use volumetric 3D ConvNet as the energy function for 3D shape pattern generation by MCMC. It is called 3D descriptor Net.

Also, the Generative Cooperative Nets (CoopNets) (Xie. PAMI 2018)[5] and (Xie. AAAI 2018) [6], which jointly trains an EBM and a generator network by MCMC teaching.

Those are the more original and earlier papers for deep EBMs with ConvNet as energy function than what you have cited, e.g., [7](Yilun Du and Igor Mordatch, 2019).

===================
References:

[1] A Theory of Generative ConvNet. Jianwen Xie *, Yang Lu *, Song-Chun Zhu, Ying Nian Wu (ICML 2016)

[2] Synthesizing Dynamic Pattern by Spatial-Temporal Generative ConvNet Jianwen Xie, Song-Chun Zhu, Ying Nian Wu (CVPR 2017)

[3] Learning Energy-based Spatial-Temporal Generative ConvNet for Dynamic Patterns Jianwen Xie, Song-Chun Zhu, Ying Nian Wu IEEE Transactions on Pattern Analysis and Machine Intelligence (TPAMI) 2019

[4] Learning Descriptor Networks for 3D Shape Synthesis and Analysis Jianwen Xie *, Zilong Zheng *, Ruiqi Gao, Wenguan Wang, Song-Chun Zhu, Ying Nian Wu (CVPR) 2018

[5] Cooperative Training of Descriptor and Generator Networks. Jianwen Xie, Yang Lu, Ruiqi Gao, Song-Chun Zhu, Ying Nian Wu. IEEE Transactions on Pattern Analysis and Machine Intelligence (TPAMI) 2018

[6] Cooperative Learning of Energy-Based Model and Latent Variable Model via MCMC Teaching. Jianwen Xie, Yang Lu, Ruiqi Gao, Ying Nian Wu. AAAI 2018.

[7] Yilun Du and Igor Mordatch. Implicit generation and modeling with energy based models. In Advances in Neural Information Processing Systems, pages 3603–3613, 2019

Thank you!

---

> ### Author Response · Authors · 2020-11-20
> **Thanks for the reminder**
>
> Hi,
>
> Thanks for the reminder about missing references to the original work.
> We have added the relevant citation in our updated manuscript.

---

### Author Response · Authors · 2020-11-20
**Updates to the manuscript**

We thank all reviewers for their constructive feedbacks.

In light of comments from reviewers, we performed further experiments on EBM to compare the learned sampler with MALA. Initially we didn’t expect 40 steps of MALA to be able to stably train the EBM until convergence. It turns out that it could, therefore comparing to the 500 steps used in [Nijkamp 2019] is no longer appropriate. We have since removed claims about saving of sampling steps from Section 5.

Longer training with MALA also shows that early during training, the learned sampler has better proposal entropy than MALA, but the advantage diminished later in the training. MALA could even be slightly better in the very late stage of training. We believe this is because the explorable volume near a sample becomes too small for the trained sampler to make a difference. We have added related discussions in Section 5.

As recommended by reviewer 3, we also tested sample quality for models trained with the two different methods. The result is the model trained with the learned sampler has higher sample quality, showing that higher proposal entropy early during training indeed helps. We have updated our paper to include this result.


Other changes to the paper includs:

1. Added citation to generative convnet paper.

2. Improve experiment description in appendix.

3. Fixed the ambiguous (misleading) description about stop gradient during the backward pass.

4. Added discussion about ergodic and aperiodic property of the proposal distribution in section 3.

5. Changed statements related to tuning step size in section 5.

6. Added Figure in Appendix that demonstrates the correctness of sampling and EBM training process.

7. Moved HMC description to section 2 and improved presentation of section 3.

8. Added comparison with gradMALA in Bayesian Logistic regression sampling task.

Thanks

Reference

Erik Nijkamp, Mitch Hill, Tian Han, Song-Chun Zhu, and Ying Nian Wu. On the anatomy of mcmc-based maximum likelihood learning of energy-based models. InProceedings of the Conferenceon Artificial Intelligence (AAAI), 2019.

---

> ### Author Response · Authors · 2020-11-25
> **Additional update to the manuscript**
>
> Following reviewer comments, we performed some additional experiment and updated the manuscript to include them:
>
> 1.  We compared ESS/s between the learned sampler and HMC, showing significantly improved performance of the learned sampler in computation speed. The result is shown in Appendix Table A2.
>
> 2. We measured FID throughout training process, which shows the FID of the replay buffer decreases more quickly when using the learned sampler, which suggests having higher proposal entropy improves the convergence of the EBM. The result is shown in Appendix Figure A3 b)
>
> Thanks

---

### Decision · Program_Chairs · 2021-01-07
**Final Decision**

**Decision:**

Reject

**Comment:**

This paper proposed an MCMC sampler that combines HMC and neural network based proposal distribution. It is an improvement over L2HMC and [Titsias & Dellaportas, 2019], with the major innovation being that, the proposed normalizing flow-based proposal is engineered such that the density of the proposal $q(x'|x)$ is tractable. Experiments are conducted on synthetic distributions, Bayesian logistic regression and deep energy-based model training.

While reviewers are overall happy about the novelty of the approach, some clarity issues have been raised in some of the reviewers' initial reviews. Also concerns on the evaluation settings, including the missing evaluation metric such as ESS/second, are also raised by the reviewers. The revision addressed some of the clarity issues, but some experimental evaluation issues still exist (e.g. comparing with L2HMC in terms of ESS/second), and the replaced MALA baseline results make the improvement of the proposed approach less clear.

I personally find the proposed approach as a very interesting concept. However I also agree with the reviewers that more experimental studies need to be done in order to understand the real gain of the approach.